# Functional Kernel Density Estimation: Point and Fourier Approaches to Time Series Anomaly Detection

**DOI:** 10.3390/e22121363

**Published:** 2020-11-30

**Authors:** Michael R. Lindstrom, Hyuntae Jung, Denis Larocque

**Affiliations:** 1Department of Mathematics, University of California, Los Angeles, CA 90024, USA; 2Global Aviation Data Management, International Air Transport Association (IATA), Montréal, QC H2Y 1C6, Canada; jungh@iata.org; 3Department of Decision Sciences, HEC Montréal, Montréal, QC H2Y 1C6, Canada; denis.larocque@hec.ca

**Keywords:** time series, anomaly detection, unsupervised learning, kernel density estimation, missing data

## Abstract

We present an unsupervised method to detect anomalous time series among a collection of time series. To do so, we extend traditional Kernel Density Estimation for estimating probability distributions in Euclidean space to Hilbert spaces. The estimated probability densities we derive can be obtained formally through treating each series as a point in a Hilbert space, placing a kernel at those points, and summing the kernels (a “point approach”), or through using Kernel Density Estimation to approximate the distributions of Fourier mode coefficients to infer a probability density (a “Fourier approach”). We refer to these approaches as Functional Kernel Density Estimation for Anomaly Detection as they both yield functionals that can score a time series for how anomalous it is. Both methods naturally handle missing data and apply to a variety of settings, performing well when compared with an outlyingness score derived from a boxplot method for functional data, with a Principal Component Analysis approach for functional data, and with the Functional Isolation Forest method. We illustrate the use of the proposed methods with aviation safety report data from the International Air Transport Association (IATA).

## 1. Introduction

Being able to detect anomalies has many applications, including in the fields of medicine and healthcare management [1,2]; in data acquisition, such as filtering out anomalous readings [3]; in computer security [4]; in media monitoring [5]; and many in the realm of public safety such as identifying thermal anomalies that may precede earthquakes [6], identifying potential safety issues in bridges over time [7], detecting anomalous conditions for trains [8], system level anomaly detection among different air fleets [9], and identifying which conditions pose increased risk in aviation [10]. Given a dataset, anomaly detection is about identifying individual data that are quantitatively different from the majority of other members of the dataset. Anomalous data can come in a variety of forms such as an abnormal sequence of medical events [11] and finding aberrant trajectories of pantograph-caternary systems [12]. In our context, we look for time series of aviation safety incident frequencies for fleets of aircrafts that differ substantially from the rest. By identifying the aircraft types or airports that have significant different patterns of frequencies of specific incidents, our model can provide insights on the potential risk profile for each aircraft type or airport and highlight areas of focus for human analysts to perform further investigations.

Identifying anomalous time series can be divided into different types of anomalous behaviour [13] such as: point anomalies (a single reading is off), collective anomalies (a portion of a time series that reflects an abnormality), or contextual anomalies (when a time series behaves very differently from most others). Identifying anomalous time series from a collection of time series, as in our problem, can be done through dimensionality reduction (choosing representative statistics of the series, applying PCA, and identifying points that are distant from the rest) and through studying dissimilarity between curves (a variant of classical clustering like kmeans) [14]. After reducing the dimension, some authors have used entropy-based methods, instead, to detect anomalies [15]. Archetypoid analysis [16] is another method, which selects time series as archetypeoids for the dataset and identifies anomalies as those not well represented by the archetypeoids. Very recently, authors have used a generalization of Isolation Forests to identify anomalies [17] and have examined the Fourier spectrum of time series and looked at shifting frequency anomalies [18]. Our approach, like Functional Isolation Forest, is geometric in flavor and we employ Kernel Density Estimation and analysis of Fourier modes to detect anomalies.

In this manuscript, we present two alternative means of anomaly detection based on Kernel Density Estimation (KDE) [19]. We use two approaches: the first and simplest considers each time series as element of a Hilbert space H and employs KDE, treating each time series in H as if it were a point in one-dimensional Euclidean space, placing a Gaussian kernel at each curve with scale parameter ξ>0. We refer to this as the **point approach** to Functional KDE Anomaly Detection, because each curve in H is treated as a point. This approach then *formally* generates a proxy for the “probability density” over H. Anomalous series are associated with smaller values of this density. This is distinct from considering a single time series as collection of points sampled from a distribution and using KDE upon points in the time series as has been done before [20]. This is a very simple, and seemingly effective method, with ξ chosen as a hyper-parameter. We also present a **Fourier approach**, which approximates a probability density over H through estimating empirical distributions for each Fourier mode with KDE. This allows us to estimate the likelihood of a given curve. Curves with lower likelihoods are more anomalous. Both methods naturally handle missing data, without interpolating. In real flight operations, sometimes it is not possible to capture and record complete information because incident data is documented from voluntary reporting, which may result in incomplete datasets. Therefore, model robustness to the impact of missing data is crucial to derive the correct understanding, which may save human lives and prevent damaged aircrafts.

The rest of our paper is organized as follows: in Section 2, we present the details and implementation of our methods; in Section 3, we conduct some experiments to investigate the strengths and weaknesses of the approaches and compare them with three other methods (Functional Isolation Forest available in Python and the PCA and functional boxplot methods available in R); following this, we apply our techniques to data from the International Air Transport Association (IATA); finally, in Section 4, we discuss our results and present some recommendations.

## 2. Functional Kernel Density Estimation

### 2.1. Review of Kernel Density Estimation

We first recall KDE over $R^{d}$, d∈N. Given a sample $S\subset R^{d}$ of *n* points from a distribution with probability density function (pdf) $f:R^{d}\to \left[0,\infty \right)$ with $\int_{R^{d}}f\left(x\right)dx=1$, KDE provides an empirical estimate for the probability density given by [19]
(1)$$\begin{array}{cc} \tilde{f}\left(x\right) & =\frac{1}{n}\sum_{y\in S}{\left|\Xi \right|}^{-1/2}K\left(\Xi^{-1/2}\left(x-y\right)\right) \end{array}$$
where Ξ is a symmetric, positive definite matrix known as the *bandwidth matrix* and *K* is a Kernel function. We choose the form of a multivariate Gaussian function so
(2)$$\begin{array}{cc} \tilde{f}\left(x\right) & =\frac{1}{n}\sum_{y\in S}\frac{e^{-\frac{1}{2}{\left(x-y\right)}^{T}\Xi^{-1}\left(x-y\right)}}{{\left(2\pi \right)}^{d/2}{\left|\Xi \right|}^{1/2}},\;x\in R^{d} \end{array}$$
and we choose [19]
(3)$$\Xi =\alpha \;\mathrm{diag}({\tilde{\sigma }}_{1},{\tilde{\sigma }}_{2},\ldots ,{\tilde{\sigma }}_{d})$$
where ${\tilde{\sigma }}_{i}$ is the sample standard deviation of the $i^{\mathrm{th}}$-coordinate of the sample points in S and
(4)$$\alpha ={\left(\frac{4}{(d+2)n}\right)}^{1/(d+4)}.$$
We used tilde (~) rather than hat (^) to denote estimators as later on we use the hats for Fourier Transform modes and wish to avoid ambiguities. In general tildes will be used for estimates derived from samples.

### 2.2. Setup, Assumptions, and Notation

The proposed methods are applicable to situations where we look for anomalous time-series relative to the sample we have. We study time series, which we consider more abstractly as being discrete samples from curves of form x(t) where x:[0,T]→R for some T>0. The space of all such curves is quite general and we limit the scope to Hilbert spaces on [0,T]. For example, we may consider spaces $H=L^{2}\left(\left[0,T\right]\right)$ or $H^{1}\left(\left[0,T\right]\right)$, the space of square integrable functions or the space of square integrable functions whose derivative is also square integrable, respectively. Within our Hilbert space, H, there is an inner product $\left(\cdot ,\cdot \right):H^{2}\to C$ and an induced norm, ||·||:H→[0,∞) where $\left||x|\right|={\left(x,x\right)}^{1/2}$. With this norm, we can define distances between elements of H.

Observations are made at *p* different times, $t_{0},$$t_{1},$ …, $t_{p-1}$ where $t_{i}=i\Delta$ with Δ=T/p and i=0,1,…,p−1. We also have $t_{p}=T$, but this time is not included in the data. Although observations are made at these times, some time series could have missing values. When a value is missing, we will say its "value" is Not-a-Number (NaN). While the set of observation points are uniformly spaced, the times at which a given time series has non-NaN values may not be.

We denote by *n* the number of time series observed, given to us as a sample of form $X={\left\{{\left\{\left(t_{j}^{\left(k\right)},x_{j}^{\left(k\right)}\right)\right\}}_{j=0}^{P_{k}-1}\right\}}_{k=1}^{n}$, where k=1,…,n indexes the time series, $P_{k}$ is the number of available (i.e., non-NaN) points for time series *k*, $0\leq t_{0}^{\left(k\right)}<t_{1}^{\left(k\right)}<\ldots <t_{P_{k}-1}^{\left(k\right)}<T$ are the times for series *k*, with corresponding non-NaN values $x_{0}^{\left(k\right)},x_{1}^{\left(k\right)},\ldots ,x_{P_{k}-1}^{\left(k\right)}\in R.$

### 2.3. Preprocessing

The methods often performed better if we normalized the data by a standard centering and rescaling. At each fixed observation time, the values of the time series were shifted to have mean zero and then rescaled to have unit variance. When the variance was already zero, the values were mapped to 0. Further remarks are given in Section 4.

Even though our methods do not assume stationary or other similar properties, applying transformations to the data before applying them can be done. For example, we may wish to make the series stationary, or to extract some characteristics (e.g., the cyclical part, or the seasonal part). This can be useful if we want to focus on finding specific types of anomalies.

### 2.4. Point Approach to Functional KDE Anomaly Detection

Our first method can be summarized as follows: treat each x∈H as a point in one dimension. Select a value for the KDE scale hyper-parameter ξ>0, and define a score functional over H by
(5)$$S_{P}\left[a\right]=\sum_{x\in X}e^{-\frac{||x-{a||}^{2}}{2\xi^{2}}},\;a\in H,$$
which, at least formally, can be thought of as a proxy to a “probability density” functional. More rigorously, one should consider measures on Hilbert spaces [21]. Assuming anomalous curves are truly rare, they should be very distant from the majority of curves and $S_{P}\left[\cdot \right]$ should be smaller at such curves. See Figure 1 for a conceptual illustration. We find that choosing ξ to be the mean of ${\left\{||a||\right\}}_{a\in X}$ to work well; another natural choice would be the median. These choices are natural because they represent a natural size/scale for the series. This approach can also be interpreted from a Fourier perspective which we remark on in Appendix A.

This method can be implemented with the following steps:Choose ξ>0.For each x∈X, compute its score from (Equation 5) where, for example, in the case of $H=L^{2}\left(\left[0,T\right]\right)$,
(6)$$||x-{a||}^{2}=\int_{0}^{T}{\left|(x\left(t\right)-a\left(t\right)\right|}^{2}dt.$$Identify anomalies as curves with the lowest score.

The integral in (Equation 6), even with some data points missing, can be computed as below:To compute $I=\int_{0}^{T}{\left|(x\left(t\right)-a\left(t\right)\right|}^{2}dt$, determine all *t*-values where both *x* and *a* are not NaN. Call these $t_{0}^{*},t_{1}^{*},\ldots ,t_{r-1}^{*}$.Define $t_{r}^{*}=T-t_{r-1}^{*}+t_{0}^{*}$, $x_{r}^{*}=x_{0}^{*}$ and $y_{r}^{*}=y_{0}^{*}$.Estimate the integral as
$$\begin{array}{cc} I & \approx \frac{1}{2}\sum_{m=0}^{r-1}\left(t_{m+1}^{*}-t_{m}^{*}\right)(|x\left(t_{m}^{*}\right)-y\left(t_{m}^{*}\right){|}^{2}+|x\left(t_{m+1}^{*}\right)-y\left(t_{m+1}^{*}\right){|}^{2}). \end{array}$$
This is a second-order accurate (trapezoidal) approximation to *I* where we have extended the signal periodically at the endpoint. This ensures that in a pathological case such as there being only a single point of observation for the integrand with value *v*, then the inner product evaluates to Tv.

### 2.5. Fourier Approach to Functional KDE Anomaly Detection

We first observe that most Hilbert spaces of interest such as $L^{2}\left(\left[0,T\right]\right)$ have a countable, orthogonal basis B={exp(2πikt/T)|k∈Z}. By considering time series as being represented by these basis vectors, we can more accurately consider a true probability density over H. In practice, we pick L∈N large and represent a∈H by
$$a\left(t\right)\approx \sum_{j=-L}^{L}{\hat{a}}_{k}e^{2\pi ikt/T}.$$
Then, up to a Fourier mode of size *L*, we can define a probability density at a∈H by $\prod_{k=-L}^{L}\zeta_{k}\left({\hat{a}}_{k}\right)$ where $\zeta_{k}$ is a pdf over C for mode *k*.

Our time series are discrete with finitely many points so we consider a Non-Uniform Discrete Fourier Transform (NUDFT). To estimate the probability density over H at *a*, we:Compute $p^{*}=\min\left\{P_{1},P_{2},\ldots ,P_{n}\right\}$.Compute the Discrete Fourier coefficients
$${\hat{x}}_{j}^{\left(k\right)}=\frac{1}{P_{k}}\sum_{m=0}^{P_{k}-1}\exp\left(-2\pi ijt_{m}/T\right)x^{\left(k\right)}\left(t_{m}\right)$$
for each k=1,…,n and for $j=0,1,\ldots ,p^{*}-1$.For each $0\leq j\leq p^{*}-1$, use KDE to estimate the pdf over C for ${\hat{x}}_{j}$, by using KDE (Equations (Equation 2)–(Equation 4)) for R or $R^{2}$ when the coefficients are all purely real/imaginary or contain a mix of real and imaginary components, respectively. Call the empirical distribution ${\tilde{\zeta }}_{j}$ for each j.For any a∈H define an estimated pdf via
(7)$$\rho_{F}\left[a\right]=\prod_{j=0}^{p^{*}-1}\zeta_{j}\left({\hat{a}}_{j}\right).$$Let the score of a∈H be
(8)$$S_{F}\left[a\right]=\log\rho_{F}\left[a\right].$$Identify anomalies in X as those whose scores given by (Equation 8) are smallest.

Due to missing data, this method does lose some information since the higher Fourier modes necessary to fully reconstruct a given time series may be discarded. Additionally, as the missing data may result in non-uniform sampling, the typical aliasing of the Discrete Fourier Transform does not take effect. In general for one of the series $x^{\left(k\right)}$, we will not have ${\hat{x}}_{P_{k}-j}^{\left(k\right)}=\bar{{\hat{x}}_{j}^{\left(k\right)}}$, where the bar denotes complex conjugation. See the remark on aliasing in Appendix B.

In multiplying the pdfs in each mode to estimate the probability density at a point in the Hilbert space, we have implicitly assumed the modes are independent. It may seem intuitive to decouple the modes by applying a Mahalanobis transformation upon the modes prior to KDE, but this results in poor outcomes. Thus, this implicit independence seems to work well in practice, without adjustments.

A Discrete Fourier transform of a signal $x_{0},x_{1},\ldots ,x_{P_{S}-1}$ measured at times ${\tilde{t}}_{0},{\tilde{t}}_{1},\ldots ,{\tilde{t}}_{P_{S}-1}$ is a representation in a new basis ${\left\{e^{\left(k\right)}\right\}}_{k=0}^{P_{S}-1}$ where $e_{j}^{\left(k\right)}=e^{2\pi ik{\tilde{t}}_{j}/T}$ for $j=0,\ldots ,P_{S}-1$. In general, such a basis of vectors for a NUDFT will not be orthogonal [22]. However, if $m=p-P_{S}\ll p$ and the $\tilde{t}$’s are a subset of a uniformly spaced set of times, we can show that the vectors are *almost* orthogonal with a cosine similarity of size O(m/p). Details appear in Appendix C. This orthogonality is not strictly necessary to run the method, but doing so allows a deeper justification of multiplying the pdfs in each mode if the Fourier modes are truly independent because the Discrete Fourier Transform is then approximately a projection onto an orthogonal basis of modes, each of which are independent.

## 3. Method Performance

We begin by illustrating the performance of our methods for some synthetic data and compare Functional KDE to other methods. The first one is the Functional Boxplot (FB) [23]. The fbplot function in the R package fda is used to obtain a center outward ordering of the time series based on the band depth concept which is a generalization to functional data of the univariate data depth concept [24]. The idea is that anomalous curves will be the ones with the largest ranks, that is, the ones that are farther away from the center. The second method is the recently proposed Functional Isolation Forest (FIF) [17], which is also depth-based and assigns a score to a curve, with higher values indicating that it is more anomalous. We used the code provided for FIF directly on GitHub [25] with the default settings given. The third is the method proposed in [26] and implemented in the R package anomalousACM [27]. This method works in three steps: (i) extract features (e.g., mean, variance, trend) from the time-series; (ii) use Principal Component Analysis (PCA) to identify patterns; (iii) Use a two dimensional outlier detection algorithm with the first two principal components as inputs. It will be referenced as the PCA method in what follows After testing them on synthetic data, we apply our techniques to real data to identify anomalies in time series for aviation events.

The methods against which we compare our methods did not have standard means of managing missing entries. For these methods, we replace missing data (NaN) in a series using Python’s default interpolation scheme. For the methods proposed in this paper, we do not have to use imputation.

### 3.1. Synthetic Data

We apply the Point and Fourier Approaches to Functional KDE, Functional Boxplot, and Functional Isolation Forest to the two scenarios described below.

**Scenario 1**: we define a base curve
$$x_{0}\left(t\right)=a_{0}\left(1+\tanh\left(b_{0}\left(t-t_{0}\right)\right)\right)+c_{0}\sin\left(\omega_{0}t/T\right),$$
with $a_{0}=5$, $b_{0}=2$, T=50, $t_{0}=T/2=25$, and $\omega_{0}=2\pi$. Ordinary curves are generated via
$$x_{0}\left(t\right)+\epsilon \left(t\right),$$
where ϵ(t) represents Gaussian white noise at every *t* with mean $\mu_{g}=0$ and standard deviation $\sigma_{g}=0.05$. We then consider a series of 7 anomalous curves:$C_{1}\left(t\right)=x_{0}\left(t\right)\left(1+r_{1}\frac{{\left(t-t^{*}\right)}^{2}}{1+{\left(t-t_{0}\right)}^{2}}\Theta \left(t-t_{0}\right)\right)+\epsilon \left(t\right)$, where $r_{1}=0.05$ and Θ denotes the Heaviside function. Thus, the function is scaled up after $t_{0}$.$C_{2}\left(t\right)=x_{0}\left(t\right)+\left(1+r_{2}\Theta \left(t-t_{0}\right)\right)\epsilon \left(t\right)$, where $r_{2}=3$. Thus, the noise is larger after $t_{0}.$$C_{3}\left(t\right)=x_{0}\left(t\right)-r_{3}\left(t-t_{0}\right)\Theta \left(t-t_{0}\right)+\epsilon \left(t\right)$, where $r_{3}=0.05$. Thus, there is a decreasing component added to the function after $t_{0}.$$C_{4}\left(t\right)=2a_{0}\Theta \left(t-t_{0}\right)+c_{0}\sin\left(\omega_{0}t/T\right)+\epsilon \left(t\right)$, i.e., the tanh is replaced by a discontinuous function.$C_{5}\left(t\right)=x_{0}\left(t\right)+E\left(t\right)$, where E(t) represents an exponential random variable at every *t* with mean 0.05.$C_{6}\left(t\right)=a_{0}\left(1+\tanh\left(2b_{0}\left(t-t_{0}\right)\right)\right)+c_{0}\sin\left(\omega_{0}t/T\right)+\epsilon \left(t\right)$, which has a slightly steeper transition rate than the base curve.$C_{7}\left(t\right)=a_{0}(1+\tanh\left(b_{0}\left(t-t_{0}\right)\right)+$$c_{0}\sin\left(\left(1+r_{7}t/T\right)\omega_{0}t/T\right)+\epsilon \left(t\right)$, where $r_{7}=0.1$ so the frequency increases with time.

Over 50 trials, we generate 70 time series, 63 normal curves, and 7 anomalous curves with each of $C_{1}$ through $C_{7}$ being used once. See Figure 2 for an illustration. We used a uniform mesh with 50 points, 0,1,…,49. Since we used a 9:1 ratio of regular to anomalous series, successful methods, after ranking curves in ascending order of “regular,” should rank anomalous curves as among the bottom 10%. We can also determine the 95th percentile for the percentile rank of each curve, to give an estimate for how much of the data would need to be re-examined to capture such anomalies. These trials can also be done by dropping data points independently at random with a fixed probability to simulate missing data. We  ran sets of trials with 0% and 10% of drop probabilities. Results for the mean percentile rank and 95th percentile of the percentile ranks are presented in Table 1 and Table 2.

**Scenario 2:** we utilized the testing examples of Staerman et al. [17]. The data consist of 105 time series over [0,1] with 100 time points. There are 100 regular curves defined by $x\left(t\right)=30{\left(1-t\right)}^{q}t^{q}$ where *q* is equi-spaced in [1,1.4]–thus there is a large family of normal curves. Then, there are 5 anomalous curves:$D_{1}\left(t\right)=30{\left(1-t\right)}^{1.2}t^{1.2}+\beta \chi_{\left[0.2,0.8\right]}$, where β is chosen from a Normal distribution with mean 0 and standard deviation 0.3 and $\chi_{I}$ is the characteristic function of *I* (there is a jump discontinuity at 0.2 and 0.8).$D_{2}\left(t\right)=30{\left(1-t\right)}^{1.6}t^{1.6}$, being anomalous in its magnitude.$D_{3}\left(t\right)=30{\left(1-t\right)}^{1.2}t^{1.2}+\sin\left(2\pi t\right)$.$D_{4}\left(t\right)=30{\left(1-t\right)}^{1.2}t^{1.2}+2\chi_{\left\{\tau \right\}}$, where τ=0.7 is a single point.$D_{5}\left(t\right)=30{\left(1-t\right)}^{1.2}t^{1.2}+\frac{1}{2}\sin\left(10\pi t\right).$

Each curve was sampled uniformly at 100 points. We did not drop any data points and, owing to the limited randomness, we only present the results of one trial. The results are presented in Table 3.

Our method is unsupervised and thus the distinction as to what constitutes an anomaly requires considering a curve’s score relative to the others and making a decision based upon this. This can involve human judgment. However, since our method returns a scalar score, we can also use a univariate outlier test on the score to formally test the hypothesis $H_{0}$: There are no anomalies. The Rosner test [28] is such a test and is available in the R package EnvStats [29]. In Appendix D, by considering scenarios where anomalies are present or absent, we show the validity of this approach.

### 3.2. Aviation Safety Reports

We now consider how our methods behave in identifying anomalous time series for aviation safety events. A discussion on method performance is deferred to Section 4.

We were provided IATA data for safety-related events of different types on a month-by-month basis from 2018–2020 for different aircraft types and airports. Aircraft types were given IDs from 1 to 64 (not every ID in the range was included). We were also given separate data pertaining to flight frequency in order to normalize and obtain event rates (cases per 1000 flights). Events of interest could include phenomena occurring during a flight such as turbulence or striking birds, or physical problems such as a damaged engine. We study two events: A and B. Event Type A is a contributing factor for one specific type of accident; Event Type B is the aircraft defense against that type of accident. To illustrate our method while preserving the confidentiality of the data, we do not state what A and B represent.

We plot histograms for the scores of Type A and B Events in Figure 3. These histograms suggest that, for events A and B, anomalous curves could be those with scores below 10 for the Point approach. Then, we consider curves anomalous by the Fourier method if they have scores below −60 for event A and −30 for event B. As the method is unsupervised, the notion of where to draw the line of being anomalous is somewhat subjective. The idea here is to raise a flag so that experts can investigate the anomalous cases more closely. The aircraft types identified as anomalous for both methods are presented in Table 4. It appears for these data, the curves deemed anomalous by the Fourier method are a subset of the curves deemed anomalous by the Point method. In Figure 4, we plot the anomalous curves (with markers) along with the normal curves (dotted lines) for the fleet IDs that were common to both approaches.

## 4. Discussion and Conclusions

### 4.1. Method Performance

From Table 1 and Table 2 with regards to Scenario 1, the Point method and FB are superior. They correctly classify $C_{1}$–$C_{3}$ and $C_{7}$ as anomalous. The Point and PCA methods significantly outperform the other methods in the more difficult $C_{4}$–$C_{6}$ curves. With these data, the Point method generally performs better without normalization. Generally all methods failed to identify the replacement of Gaussian white noise with exponential noise ($C_{5}$) as anomalous, although the un-normalized Point approach succeeded. Additionally, all methods considered, except PCA, had difficulty identifying the discontinuous replacement of the hyperbolic tangent ($C_{4}$) and a slightly steeper hyperbolic tangent ($C_{6}$). The PCA method was not as good as the others for identifying the noise increases ($C_{2}$) and the frequency increase with time ($C_{7}$). This suggests that not all methods are effective at detecting the same types of anomalies and that they may be complementary.

From Table 3 for Scenario 2, the Fourier approach with normalized data, FIF on un-normalized data, and PCA classify equally correctly. Data can always be normalized and this is therefore not a problem for the Fourier method. In this example, the Point approach fares better with normalization. However, this method and the FB method are not as effective as the FIF, PCA, and Fourier methods.

From our experiments, when there was a large family of curves as with Scenario 2, the Fourier method performed better at detecting anomalies, especially when provided normalized data. But when the family of curves were all close to the same, except for noise, the Point method was better, with or without normalization. Providing more theoretical understanding as to whether these are general phenomena is left for future work.

### 4.2. Aviation Safety Data

From Figure 4, it appears the methods can detect different sorts of anomalies. In the case of Type A events, the anomalous curves appear to have anomalously large values at an isolated point or over small range of values. The anomalies in Type B events are more interesting and subtle. Even some of the normal curves have sizeable event frequencies, sometimes even exceeding the anomalous curves. But on the average it seems the anomalous curves are higher. In the case of curve 9, the reason it is deemed anomalous is not immediately intuitive. Whether such differences are of a concern to safety would require follow-up from safety inspectors.

To prevent aviation accidents, identifying the potential hazards and risks before they evolve into accidents is the key to proactive safety management. While collecting and analyzing data manually is a time-consuming process, especially on a global scale, the risk identification process may remain reactive process if there is not an automated process. The application of the anomaly detection will enable proactive data-driven risk identification in global aviation safety, by continuously monitoring aviation safety data across multiple criteria (e.g., airport, aircraft type and date), then automatically raising a flag when the model detects any anomalous patterns.

The proposed model shows potential value in automatically detecting potential risk areas with robustness from missing data; however, the interpretation of the model still requires future study. As safety risk is an outcome of complex interactions between multiple factors, including human, machine, environment, and other hidden factors, understanding the full context of such risk requires in-depth investigation and validation from multiple experts. While the model can identify some anomalous patterns, this does not take into consideration the interactions. For example, some aircraft fleets fly more frequently over certain pathways than others. Thus, some differences identified as anomalous due to aircraft type may actually stem from location. Therefore, there will always be a human layer between the model and the interpretation of the model.

### 4.3. Comments on the Models

There are various degrees of freedom the proposed methods allow for, which are worth noting. Firstly, the point method could be generalized to compute $H^{1}\left(\left[0,T\right]\right)$, and higher Sobolev norms too, but that could lead to additional hyper-parameters in how heavily to weigh the derivative terms. With the Fourier approach, it may seem more appropriate to replace the NUDFT with a weighted combination of terms that more accurately reflects the non-uniform spacing, i.e., a Riemann Sum. Interestingly such an approach tends to make the results slightly worse, hence our choice to use the standard NUDFT.

We anticipate these methods perform well when the time series are sampled at regular intervals and a small portion of entries are missing. If the number of missing entries is very large, this makes inner products computed with the Point method less accurate (without additional interpolations) and the preprocessing of shifting and rescaling could result in poorer outcomes due to a limited sample size upon which to base the normalizations. For many applications, however, most data are present.

### 4.4. Future Work

We note that our proposed methods aim to identify anomalous time series relative to the sample taken. In general, even if all time series are sampled from the same distribution, due to low probability events, some time series could still be anomalous relative to the sample given. As such, our work has mostly been an empirical investigation of the methods; however, by adding further assumptions on the underlying distribution of time series, it could be possible to obtain a more theoretical basis for the method performance. This would be worth investigating, but is beyond our current work. Going hand-in-hand with this theory it would be interesting to investigate the optimal choice of ξ in the point approach, to understand how the Fourier modes being treated as independent works as effectively as it does, or to more rigorously establish classes of problems when the Point or Fourier approaches are superior.

In conclusion, we have presented two approaches to detecting anomalous time series using KDE to generate functionals to score a series for its degree of anomalousness. The methods handle missing data and perform well in comparison to other methods.   

## Figures and Tables

**Figure 1 entropy-22-01363-f001:**
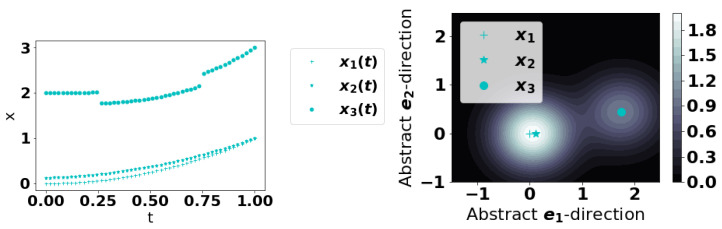
A visual depiction of the Point method. The curves are time series in a Hilbert space H but after applying KDE, there is a score associated to each point in H. In the cartoon, curves 1 and 2 are similar and curve 3 is anomalous. (**Left**): the time series. (**Right**): a representation of them with associated scores in the color scale. In reality, the space is infinite dimensional and this is only a conceptual illustration.

**Figure 2 entropy-22-01363-f002:**
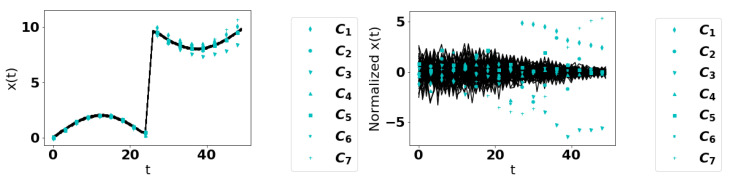
Plot of 63 normal curves and the 7 anomalous curves $C_{i}\left(t\right)$, i=1,…,7. Left: un-normalized. Right: normalized.

**Figure 3 entropy-22-01363-f003:**
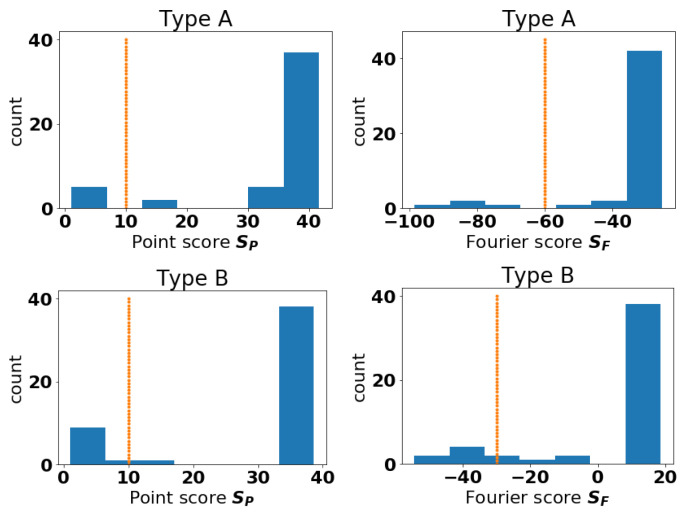
Histogram of scores for Point and Fourier methods for Type A Point (**top-left**), Type A Fourier(**top-right**), Type B Point (**bottom-left**) and Type B Fourier (**bottom-right**). The dashed vertical line represents the division we chose between anomalous (left of line) and normal (right of line). The Sturges estimate was used to set bin widths [30].

**Figure 4 entropy-22-01363-f004:**
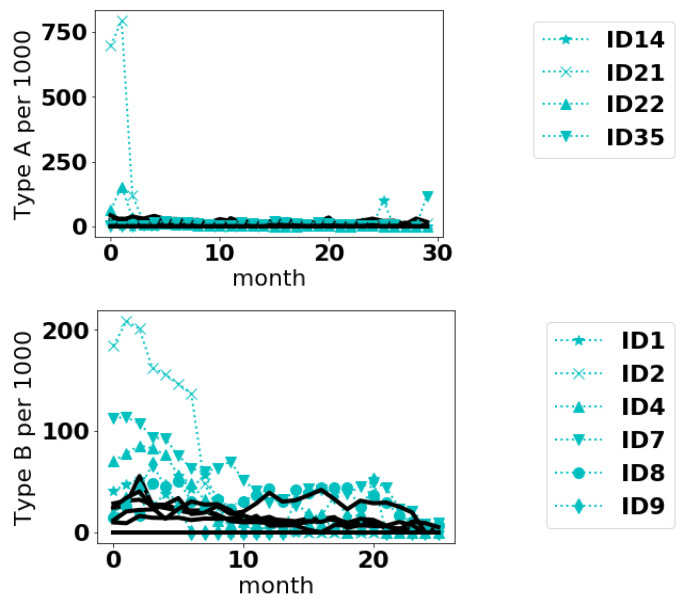
Plots of the time series for Type A and Type B events. Anomalous are dotted curves with markers in the legend; normal curves are solid black curves.

**Table 1 entropy-22-01363-t001:** Mean percentiles (out of 100) for curves $C_{1}$–$C_{7}$ in Scenario 1. A correct classification is a percentile less than or equal to 10 (in bold in the table). The -*N* suffix denotes the data were normalized by the pre-processing described in Section 2.3; the -*U* suffixed denotes the data were un-normalized. Note that method FB is not affected by the normalization.

Method	Lost	$C_{1}$	$C_{2}$	$C_{3}$	$C_{4}$	$C_{5}$	$C_{6}$	$C_{7}$
Point-N	0%	**4.3**	**5.8**	**1.4**	21	12	24	**2.9**
Point-U	0%	**4.3**	**5.7**	**1.4**	14	**8.8**	16	**2.9**
Fourier-N	0%	**4.7**	**1.9**	**2.8**	28	51	29	**8.5**
Fourier-U	0%	**5.8**	**4.0**	**2.8**	38	43	35	**1.7**
FIF-N	0%	51	24	72	56	79	58	13
FIF-U	0%	19	**2.2**	**4.8**	18	53	19	**5.3**
PCA	0%	**7.5**	20.	**4.3**	**7.5**	53	**9.4**	11
FB	0%	**4.5**	**5.6**	**1.8**	36	21	37	**2.5**
Point-N	10%	**4.3**	**6.0**	**1.4**	23	18	29	**2.9**
Point-U	10%	**4.3**	**5.7**	**1.4**	20.	14	23	**2.9**
Fourier-N	10%	**4.3**	**4.5**	**2.5**	28	43	36	**4.0**
Fourier-U	10%	45	59	50	46	49	53	49
FIF-N	10%	50.	21	75	48	74	51	13
FIF-U	10%	45	15	29	42	48	44	32
PCA	10%	32	20.	**6.1**	36	47	35	**7.7**
FB	10%	**7.5**	**8.7**	**4.2**	47	24	49	**5.1**

**Table 2 entropy-22-01363-t002:** The 95th percentile of the percentile ranks (out of 100) for curves $C_{1}$–$C_{7}$ in Scenario 1. See Table 1 caption for -*N* vs. -*U* distinction.

Method	Lost	$C_{1}$	$C_{2}$	$C_{3}$	$C_{4}$	$C_{5}$	$C_{6}$	$C_{7}$
Point-N	0%	4.3	6.5	1.4	47	42	62	2.9
Point-U	0%	4.3	5.7	1.4	30.	12	46	2.9
Fourier-N	0%	8.6	3.6	4.3	67	99	74	13
Fourier-U	0%	5.7	4.3	4.2	84	94	77	2.9
FIF-N	0%	84	69	92	90	100	96	30
FIF-U	0%	57	7.5	11	55	97	53	11
PCA	0%	25	67	7.1	17	96	27	47
FB	0%	5.7	5.7	2.9	75	74	75	2.9
Point-N	10%	4.3	7.1	1.4	52	51	72	2.9
Point-U	10%	4.3	5.7	1.4	46	43	59	2.9
Fourier-N	10%	6.5	10.	5.1	61	91	89	5.7
Fourier-U	10%	84	97	92	93	93	92	89
FIF-N	10%	81	67	97	91	99	95	39
FIF-U	10%	84	27	56	88	95	94	56
PCA	10%	82	60.	31	73	94	73	19
FB	10%	11	13	10.	75	75	75	9.4

**Table 3 entropy-22-01363-t003:** Percentiles (out of 100) for curves $D_{1}$–$D_{5}$ in Scenario 2. A correct classification is a percentile less than or equal to 4.8 (in bold in the table) since 5/105=4.8%. See Table 1 caption for -*N* vs. -*U* distinction.

Method	$D_{1}$	$D_{2}$	$D_{3}$	$D_{4}$	$D_{5}$
Point-N	74	**0.95**	6.7	73	**1.9**
Point-U	83	**0.95**	44	85	71
Fourier-N	**3.8**	**4.8**	**1.9**	**2.9**	**0.95**
Fourier-U	**1.9**	8.6	30	**0.95**	**2.9**
FIF-N	**1.9**	28	**3.8**	10	**0.95**
FIF-U	**1.9**	**2.9**	**3.8**	**4.8**	**0.95**
PCA	**4.8**	**2.9**	**3.8**	**1.9**	**0.95**
FB	75	**0.95**	21	75	75

**Table 4 entropy-22-01363-t004:** IDs of anomalous flights for events A and B. Columnwise, the bolded IDs are common to both methods for a given event type.

Method	Type A	Type B
Point	**14**, **21**, **22**, 23, **35**	**1**, **2**, **4**, 5, 6, **7**, **8**, **9**, 25, 34
Fourier	**14**, **21**, **22**, **35**	**1**, **2**, **4**, **7**, **8**, **9**

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
