# Peer review of "Functional Kernel Density Estimation: Point and Fourier Approaches to Time Series Anomaly Detection"

_entropy, 2020, doi:10.3390/e22121363_

Round 1

Reviewer 1 Report

The authors developed two unsupervised methods of anomaly detection based on Kernel Density Estimation. In my view, the suggested methods based on Kernel Density Estimation are formally correct and can be of interest for the readers of the journal ENTROPY.

I consider the topic of the paper to be interesting for potential readers of ENTROPY. I suggest accepting the manuscript for publication.

Reviewer 2 Report

This paper presents an unsupervised method to detect anomalous time series among a collection of time series. It is interesting and well-written. I think it deserves to be published.

Reviewer 3 Report

I feel that the revision and response have addressed my concerns on the paper. The addition of the Rosner test on the score, I believe, added the necessary completeness of the results. Hence, I would recommend its publication. 

This manuscript is a resubmission of an earlier submission. The following is a list of the peer review reports and author responses from that submission.

Round 1

Reviewer 1 Report

The paper is focused on the problem of detecting anomalous time series among a collection of time series. The authors developed two alternative methods of anomaly detection based on Kernel Density Estimation: the point approach which formally generates a proxy for the probability density over the considered Hilbert space, and also the Fourier approach, which approximates a probability density over the Hilbert space through estimating empirical distributions for each Fourier mode with KDE. This allows estimating the likelihood of a given curve. Curves with lower likelihoods are considered here more anomalous.

According to the authors, the suggested approaches are unsupervised. The proposed methods are based on pure heuristics. I am missing a more detailed model specification of the considered time series. In fact, the authors consider multivariate methods based on p-variate vector observations. Moreover, the notion of the anomalous time series is rather vague. There is no formal specification. In my view, the suggested methods based on Kernel Density Estimation are formally correct and can be of interest for the readers of the journal ENTROPY. However, the presented problem of anomaly detection and the suggested methods are based mainly on data manipulation methods, without transparent model-based approach. There is no comment on stationarity or other model assumption of the considered time series and or the (auto)covariance structure (the considered examples are based on simple p-dimensional vectors with independent noise).

To conclude, I consider the topic of the paper to be interesting for the potential reader of ENTROPY. However, the methods are too heuristic to my taste. On the other hand, for their money, the authors have the right to publish the results of this type.

Reviewer 2 Report

In this paper, the authors proposed two time series anomaly detection approaches for samples from a Hilbert space, the point approach and the Fourier approach. Then the methods are illustrated against both synthetic and real aviation data. While there are certainly innovative elements in the two approaches proposed, I feel that the analysis and data experiments are not sufficient.

  1. In both approaches, the data points with lowest score are identified as an anomalies. This has the potential of falsely identifying normal data points as anomalies. The authors never seem to address this concern. In the data analysis, the proposed approaches should be run against data sets with no anomalies, see how they perform. 
  2. In general, there is no theoretical analysis of these two approaches to quantify their performance. In addition, they have not been compared to any of other methods. 

As pointed out above, the two proposed approaches need more analysis to justify that they can produce reasonable outputs, and demonstrate pro and con comparing to other existing methods. Without these analysis, it can not judged that whether these approaches can be adapted in any impactful data analysis. Based on this assessment, I feel this paper should not be published in the current form.